# ODNS Clustering: Unveiling Client-side Dependency in Open DNS Infrastructure

## ABSTRACT

There are over a million open DNS servers in the wild. However, not all servers perform recursive queries directly. Instead, many DNS forwarders forward queries to upstream recursive servers or other DNS forwarders for name resolving on their behalf. The groups of open servers that have such dependencies on each other form **ODNS Clusters**. The dependencies can result in vulnerabilities; yet we have little knowledge of the ODNS cluster structure. In this work, we measure the inter-dependence of open DNS resolvers and find that 1.9 million open DNS servers form *only* 81,636 ODNS clusters. We further analyze the characteristics of the clustered ODNS structure. The key observations include biased cluster size distribution, discrepancy of ODNS infrastructures among countries, concentration in major public DNS server providers, and potential security and resilience risks due to the dependence.

## 1 INTRODUCTION

The Domain Name System (DNS) serves as a foundational infrastructure for the web [16, 25, 44], facilitating the translation of human-readable domain names into machine-readable IP addresses. Open DNS infrastructure (ODNS) provides a free entrance for billions of web users to access DNS services. There are more than one million IP addresses hosting open DNS servers in the wild [23, 34, 35]. However, not all of these open DNS servers issue queries to authoritative name servers. Specifically, DNS forwarders [42] do not resolve domain names by themselves. Instead, they forward queries from clients to another upstream server, such as a public DNS server or a dedicated gateway. On the client side, the multi-layer forwarding dependency between forwarder and upstream servers results in confusing hierarchies or unexpected dependencies. For instance, multi-level forwarding chains or loops may pose potential security risks [46].

Forwarding dependencies in ODNS infrastructure means that a large number of forwarders *may* in fact rely on the responses of a small set of upstream servers [20]. We refer to the collection of upstream servers and forwarders that have direct (or indirect) dependencies as an **ODNS cluster**. As a result, open DNS servers form multiple ODNS clusters with dependencies and naturally reveal the status of DNS infrastructure. Yet we have little knowledge of the ODNS cluster structure, such unclear clustered dependencies can amplify vulnerabilities and the impact of malicious attacks [38, 48]. It therefore becomes difficult to identify critical points of failure [6, 9], as seemingly distinct servers may actually be in the same ODNS cluster. Alternative DNS servers set by users without prior knowledge may be universally affected by the same upstream server and lead to a failure of the redundancy mechanism.

Consequently, we argue that understanding the client-side dependencies of open DNS servers is vital for improving DNS configurations and facilitating a better understanding of DNS infrastructure. In this work, we therefore measure the dependencies of open DNS

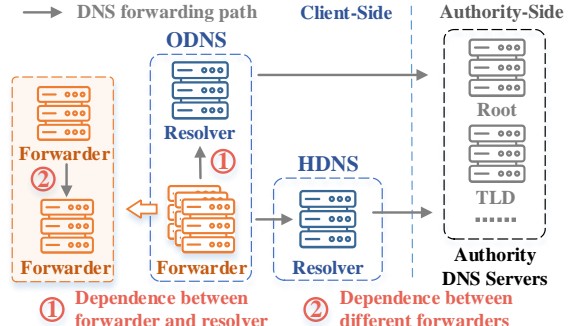

**Figure 1: DNS infrastructure and client-side structure.**

servers and divide open DNS servers into multiple ODNS clusters. To this end, we propose an ODNS clustering method that leverages the footprints left in caches during the forwarding process to identify ODNS clusters. We further propose cluster aggregation to improve clustering accuracy for large public DNS servers that have complex infrastructure [37, 40]. Besides, we propose a forwarder classification method based on forwarding behaviors inside clusters, revealing the composition of large ODNS clusters.

Overall, we identify 81,636 distinct ODNS clusters for 1.9 million open DNS servers in the wild. We further analyze the characteristics of the cluster and make the following key observations[1]:

- A significant portion (95%) of open DNS servers rely on other servers for name resolution. Moreover, we see a heavily biased distribution of cluster size, where 0.25% top clusters cover 44.1% of open DNS servers.
- Cluster size distribution varies significantly across countries. Some countries are dominated by large clusters, indicating higher dependency (*e.g.,* China), while others exhibit a more balanced distribution (*e.g.,* US and FR). This implies differences in DNS infrastructure across countries.
- ODNS clusters that are led by major public DNS servers cover 47% of the open DNS servers. The use of anycast results in many clusters for one public DNS with unbalanced cluster size distribution. This serves as another evidence of infrastructure concentration [12, 31].
- Over 9% of the ODNS servers exhibit a range of misconfigurations or malicious behavior as they direct web requests to potentially harmful destinations. The dependence captured by clusters amplifies the impact of such behavior.
- ODNS cluster consists of forwarders with diverse behaviors. Notably, about 61.7% forwarders are non-caching proxies. These proxies may be leveraged by attackers to attack the upstream resolvers within clusters.

---

[1]We make the data and results publicly available in [8].

## 2 BACKGROUND AND RELATED WORK

### 2.1 Client-side DNS Struture

The client-side DNS infrastructure consists of multiple layers of servers. Previous works have investigated the hierarchical structure of DNS [24, 26, 29, 30, 33, 42]. The key components of the DNS infrastructure are illustrated in Fig. 1. We refer to servers that accept requests directly from *any* client as the *open DNS servers* (**ODNS**), while servers that cannot be directly accessed by clients are *hidden DNS servers* (**HDNS**). Functionally, servers within the ODNS can be categorized into *forwards* and *resolvers*. Forwarders forward the original DNS query either to an upstream forwarder or a designated egress resolver, while the egress resolver ultimately communicates with the *authoritative DNS servers* (**ADNS**). The forwarding behavior in the client-side causes the dependency between open DNS servers, including ① the dependency between the forwarder and resolver, and ② the dependency between different forwarders.

Prior studies have focused on identifying forwarders and resolvers by correlating initial queries with logs from ADNS. Luo *et al.* [27] and Xu *et al.* [45] focus on matching forwarders and resolvers by encoding details of the forwarder in the request domain name in order to identify the corresponding resolver for the requested forwarder. Nawrocki *et al.* [33] and Censys [1, 13] establish the linkage between forwarders and resolvers by encoding resolver IP within ADNS responses. While these approaches can discover the dependency between the upstream backend resolver and open forwarders, they fail to illustrate the complete client-side structure of DNS infrastructure. Firstly, ADNS can only record DNS queries that do not hit the client-side cache [31], meaning it cannot unveil the hidden dependencies on the client side (*e.g.,* the dependencies between forwarders). Secondly, existing works primarily measure and analyze well-known public DNS servers, relying on prior knowledge of backend server addresses [45], and lack a universal methodology for measuring all ODNS servers comprehensively.

### 2.2 ODNS Clusters

An important observation about the above setup is that multiple DNS forwarders may rely on the same upstream DNS server. We refer to this collection of an upstream DNS server and the forwarders that have direct (or indirect) dependencies on it as an ***ODNS cluster***. Forwarders in an ODNS cluster may forward queries directly to a resolver (direct), or forward through other forwarders with multiple hops (indirect). All open DNS servers can be divided into multiple ODNS clusters due to the limited number of egress resolvers. It is worth noting that a server may belong to multiple clusters due to anycast or forwarding strategies (discussed in Section 3.2).

Because the forwarding strategy of open DNS servers is typically configured by an unknown third party (*e.g.,* Router administrator [2]), the dependencies in client-side ODNS infrastructure are invisible to clients and the ADNS administrators. Such unknown dependencies may cause potential security risks:

***Invalid redundancy configuration:*** The dependence of DNS infrastructure makes the redundancy configuration of DNS resolvers invalid. Specifically, without prior knowledge of the interdependencies of open DNS servers, configuring primary DNS and

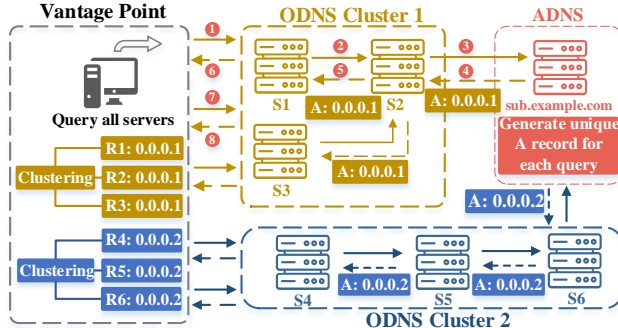

**Figure 2: ODNS dependency detection and clustering.**

secondary DNS which belong to the same ODNS cluster leads to a higher risk of a single point failure.

***Amplify the impact of malicious responses:*** The impact of malicious behaviors (*e.g.,* hijacking, cache poisoning) may be amplified by ODNS cluster. Because malicious behavior against the upstream resolver can further affect the clients of forwarders. Although the query processing of forwarders is not directly compromised, they obtain malicious responses from the cache of upstream resolvers.

***Exposing vulnerable entrances for attackers:*** Queries from forwarders in ODNS cluster are forwarded to upstream servers or ADNS, which means that some poorly configured forwarders may be used by attackers as entrances to attack upstream forwarders. Larger clusters suffer from more risks as they contain more uncontrollable attack vectors (exploited forwarders).

We argue that unveiling the client-side dependencies with ODNS clustering can help users achieve better DNS configuration and help administrators improve management. We widely measure the dependence of ODNS servers and take a further step toward understanding the client-side structure of ODNS infrastructure.

## 3 METHODOLOGY

To measure the dependencies among all open DNS servers, our goal is to let servers with dependencies demonstrate consistent response behavior, while others reply with different responses.

### 3.1 ODNS Clustering

The underlying idea for measuring the dependencies is straightforward: Servers with dependencies should retrieve the same cached record directly from the *same* upstream server if this domain has been previously accessed. Thus, we leverage the cache of upstream DNS servers to label different clusters by responding with unique record content for each query from a controlled ADNS. Fig. 2 illustrates our measurement method and an example scenario. In our example scenario, we see that servers $S1$-$S3$ have dependencies, and $S4$-$S6$ have dependencies. Our method tries to detect the dependencies in $S1$-$S6$ and divides these servers into two ODNS clusters ($S1$-$S3$ and $S4$-$S6$). The whole process involves ODNS discovery, ODNS labeling, and ODNS clustering, which are detailed below.

***Step 1: ODNS discovery.*** We first acquire a list of all open DNS servers. To this end, we use Zmap-based [14] script to send DNS A queries to all routable IPv4 addresses with an unused domain newly

registered by us. An IP is considered to be an open DNS server if it responds with a NOERROR reply code and has an A record. Through this step, we obtain a complete list of open DNS servers. Meanwhile, we check the transparent forwarders [33] which can be utilized as additional vantage points.

**Step 2: ODNS labeling.** We next leverage the footprint in the cache for labeling all ODNS servers. For this, we built a controlled ADNS for the domain (denoted as *sub.example.com* here for illustration propose) we registered. Note, this domain must not have been used before this step. Specifically, our vantage point sends A queries for *sub.example.com* to each server in the open DNS server list compiled in the previous step. For example, the vantage point will first send the query to $S1$. Ultimately, the query will be responded by our controlled ADNS. For each A query received, ADNS generates a unique A record (*e.g.,* 0.0.0.1). This unique A record will be returned to $S1$ through the forwarding chain and be cached due to the ubiquitous cache structure in resolvers [30]. The vantage point will then send the query to the next server on the list. For instance, the next query will be sent to $S2$. In this example, it will directly hit the cache in $S2$, and return the same (unique) A record. The cached responses act as the **labels** for ODNS servers.

**Step 3: ODNS clustering.** Finally, we use the shared A records (the label we get in previous steps) to group servers with the same label into a single cluster. This is again illustrated in Fig. 2. Here, the DNS queries to $S1$-$S3$ traverse through $S2$ to reach the ADNS. Consequently, they will receive the same A record, which is cached (0.0.0.1). Similarly, queries to $S4$-$S6$ will reach ADNS via $S6$. Given that our ADNS server delivers unique responses for each query, $S6$ obtains the A record 0.0.0.2. Thus, we can easily divide the ODNS address space into two ODNS clusters based on the A record responses and identify the client-side structure of ODNS infrastructure.

## 3.2 Public DNS Clusters

As public DNS resolvers become a common default configuration, our measurement methods encounter challenges when addressing clusters formed by public DNS resolvers, primarily due to their use of multiple Points-of-Presence (PoPs) with anycast addresses and fragmented backend caches [11, 37]. Fig.3 illustrates how public DNS resolver behaves during the clustering process. For forwarders specifying a public resolver (with anycast address) as the upstream server, queries are routed to the nearest PoP and then handled by one fragmented backend cache component. Consequently, even if a domain was recently resolved by a public DNS resolver, subsequent queries for the same domain may miss the previous cache. This results in forwarders being divided into more clusters unexpectedly.

We now describe our approach for handling multiple ODNS clusters formed by public resolvers. We aggregate clusters created by volatile dependencies (*e.g.,* randomly assigned fragmented caches) while preserving clusters formed by stable dependencies (*e.g.,* multiple PoPs with anycast). To this end, we perform multiple rounds of ODNS clustering using different controlled subdomains and obtain multi-round clustering results. The core of our approach is:

① **Cluster aggregation for fragmented caches:** In multi-round measurements, servers in such clusters may be reallocated to one another in different rounds (*e.g.,* changes in cache allocation of the public resolver). We leveraged the shifts across multiple rounds

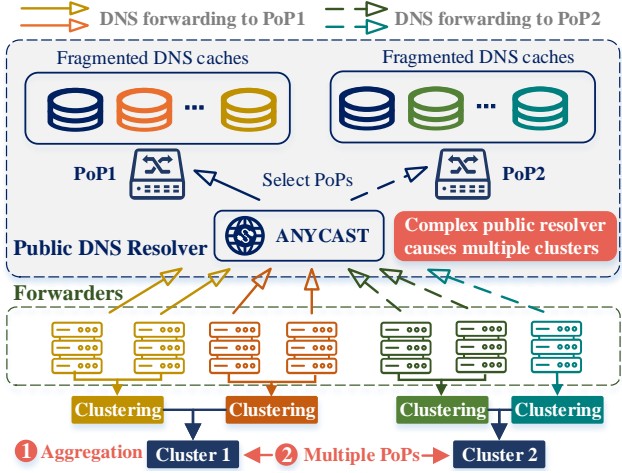

**Figure 3: Clusters with public resolvers.**

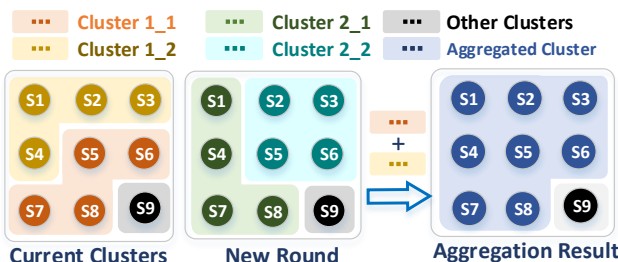

**Figure 4: Example for cluster aggregation.**

to aggregate clusters. We provide a quick example of the aggregation process in Fig. 4. Among them, servers $S1$-$S9$ represent ODNS servers in the ODNS space, which are divided into multiple clusters during the clustering process. Initially, we have a clustering result (donated as Current Clusters) consisting of three clusters: clusters1_1, clusters1_2, and clusters1_3. In the clustering result in another detection round, we observe that the fragmented cache's random allocation has generated a different clustering result (donated as New Round) of clusters1_1 and clusters1_2 to clusters2_1, clusters2_2. We found that across multiple rounds of clustering, servers $S1$-$S8$ will generate variable clustering results, but never mixed with unrelated clusters (donated as $S9$). By aggregating the clusters1_1 and clusters1_2, we can achieve a more stable clustering result. The specific aggregation method and principles are detailed in Algorithm 1.

Algorithm 1 outlines the workflow of cluster aggregation. The current clustering result is denoted as $C_{now}$, while the clustering result from the new round is denoted as $C_{new}$. The goal is to use $C_{new}$ to merge clusters in $C_{now}$ that may be related. For each cluster in $C_{new}$ (lines 1-2), we calculate the overlap ratio with each cluster in $C_{now}$ (lines 4-6). If the overlap ratio exceeds a predetermined threshold $\alpha$ (lines 7-9), the clusters are considered related, and we merge them (lines 10-13). The threshold[2] $\alpha$ represents the

---

[2] $\alpha$ is set to 0.1 in our measurements because existing work [30] has shown that the number of fragmented caches is usually less than 10.

**Algorithm 1:** Cluster Aggregation

---

**Input:** Current clustering result $C_{now}$ and new round $C_{new}$
**Output:** Aggregation result $C_{now}$

**1 for** *Each $Key_{new} \in C_{new}.keys()$* **do**
**2**     $Tags$ = []
**3**     // Traverse keys of all clusters
**4**     **for** *Each $Key_{now} \in C_{now}.keys()$* **do**
**5**        $Overlap = C_{now}[Key_{now}] \cap C_{new}[Key_{new}]$
**6**        $Overlap_{rate} = Overlap.size()/C_{new}[Key_{new}].size()$
**7**        **if** *$Overlap_{rate} \geq \alpha$* **then**
**8**           $Tags.append(Key_{now})$
**9**        **end**
**10**        **for** *Each tag $\in Tags[1:]$* **do**
**11**           $C_{now}[Tags[0]] = C_{now}[Tags[0]] \cup C_{now}[tag]$
**12**           $C_{now}.pop(tag)$
**13**        **end**
**14**     **end**
**15 end**
**16** return $C_{now}$

---

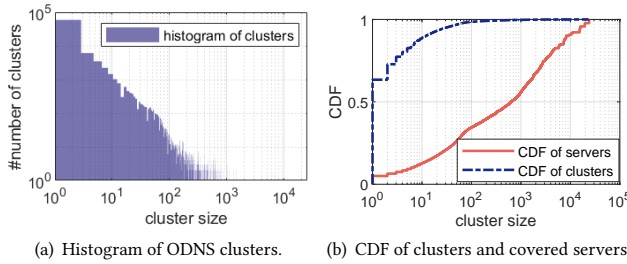

(a) Histogram of ODNS clusters.    (b) CDF of clusters and covered servers.

**Figure 5: Distribution of cluster size and open DNS servers.**

probability that servers in the current cluster will remain in the same fragmented cache in the next round of detection.

② ***Clustering for multiple PoPs:*** Because of the use of anycast address [37], public resolvers will also form multiple clusters due to multiple PoPs. Unlike the formation of fragmented caches above, we do not need to further aggregate this stable cluster structure. This is because the queries from forwarders will be forwarded to the nearest PoP and will not be changed frequently in future measurements. As such, a public DNS server can correspond to multiple clusters. For DNS servers with multiple PoPs sharing an anycast address (*e.g.,* public DNS), our method can naturally divide forwarders that depend on PoPs into different clusters.

### 3.3 Vantage Points and Transparent Forwarder

Revealing that an anycast IP is in different clusters requires vantage points corresponding to the PoPs service area. We deployed 5 controlled vantage points in Singapore, the United States, Ireland, Ukraine, and Brazil for ODNS discovery and labeling. However, it is still insufficient to measure all PoPs behind public resolvers. To this end, we turned our attention to transparent forwarders [33]. Transparent forwarders respond to clients using the address of the upstream server instead of itself. Such behavior exposes its upstream server to clients, and this upstream server is usually a public DNS

resolver. We leverage about 370k transparent forwarders across 186 regions as additional vantage points for measurement.

## 4 MEASUREMENTS & ANALYSIS

We conducted measurements with 5 controlled subdomains and repeated queries 5 times in each run. We show the cluster-level measurement results and findings in this section.

### 4.1 ODNS Clusters In-the-Wild

We find over 1.9 million addresses of open DNS servers in the wild. Among them, 972,383 servers complete the measurement process, the churning of IPs is caused by some open DNS devices that change addresses in a short period (*e.g.,* servers with DHCP [36]). These volatile addresses [18, 23] are not within the scope of our research. Our method groups open DNS servers with successful responses into 81,636 clusters, distinguished by A record responses. We illustrate the distribution of ODNS cluster size in Fig. 5(a) and the cumulative distribution function of cluster size and open DNS servers covered by different cluster sizes in Fig. 5(b).

The size of ODNS clusters ranges from only a few DNS servers to thousands. The size of the ODNS cluster demonstrates features of the DNS infrastructure. Our results reveal that 48,894 clusters contain only one IP address (covering 5% of servers after eliminating duplicate IPs). Such clusters with only one DNS server signify an individual resolver. Notably, the top 207 clusters (0.25%) with a size of over 1,000 contain over 429,499 open DNS servers in total, accounting for 44.1% of the open DNS servers. These ODNS clusters typically come from public DNS or gateways with great popularity.

> ***Observation 1:*** 95% open resolvers exhibit dependencies on others for name resolution as they fall into clusters with more than 1 server. Notably, the distribution of cluster size is heavily biased with 0.25% top clusters containing 44.1% open DNS servers.

### 4.2 ODNS Clusters with Countries

Forwarders typically opt for a nearby upstream resolver to improve service quality and reduce latency [43]. Consequently, we hypothesize that servers in the same ODNS clusters are in the same region. We therefore explore the geographical patterns of the ODNS clusters. For this, we map all ODNS IP addresses to their country codes using IPINFO [4]. In cases where an ODNS cluster is composed of servers from multiple countries, we use the country with the highest proportion to represent cluster geographical identity. We present the proportion of servers from the country with the highest proportion within each cluster. Fig. 7 illustrates the cumulative distribution function of the proportion of servers, revealing that over 95.5% of clusters have servers 99% from the same country. This result confirms the concentrated geographical distribution among servers in the same ODNS cluster.

We analyze the cluster sizes of each open DNS server belonging in each country. Fig. 6 illustrates countries with the most open DNS servers. Results show significant differences in the distribution of clustering size, which implies regional disparities of DNS infrastructures. In countries like China and South Korea, most DNS servers

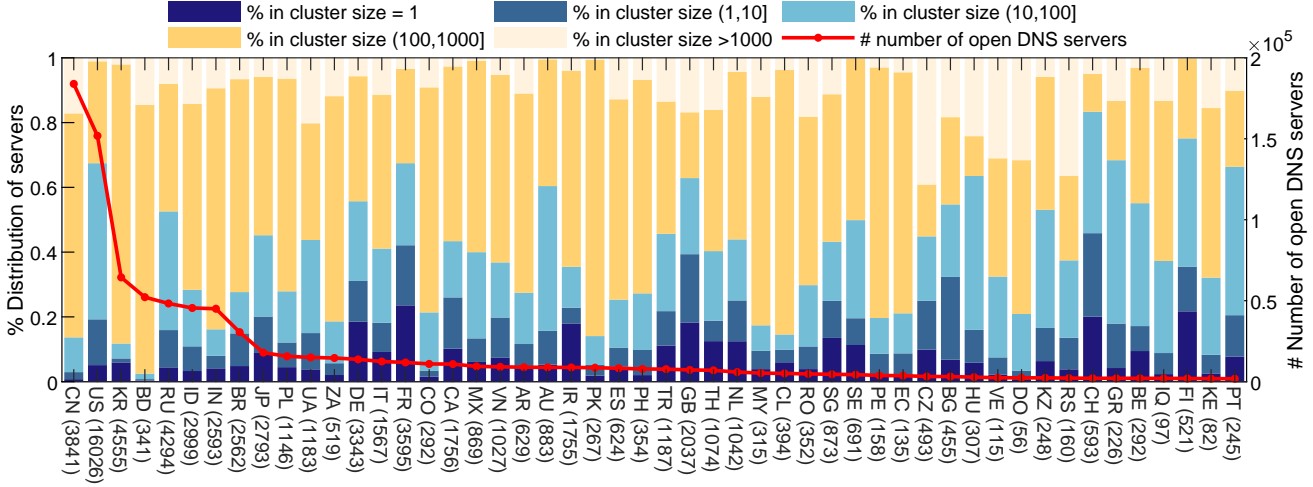

**Figure 6: Top 50 countries with the most open DNS servers.**

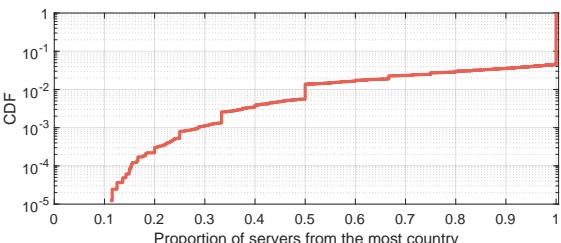

**Figure 7: CDF of servers' proportion from the most country in ODNS cluster. Note that the $y$-axis is in log scale.**

**Table 1: Clusters for top public DNS.**

| Provider | # of clusters | % covered rate |
|---|---|---|
| Google | 268 | 27.99% |
| Cloudflare | 228 | 9.76% |
| OpenDNS | 46 | 5.33% |
| Yandex.DNS | 118 | 4.24% |
| Others | — | 4.63% |

### 4.3 ODNS Clusters with Public DNS

The above shows that often public DNS acts as an upstream DNS provider within an ODNS cluster. We next try to understand the impact of public DNS within their ODNS clusters.

***Client-side centralization of Public DNS.*** Concern about DNS centralization has been mounting over the past few years [12, 27, 31]. We use the list in [5] to obtain 28 popular public DNS service providers. To discover the popularity of forwarders on public DNS, we identify ODNS clusters containing these well-known public DNS service IP addresses. We summarize the cluster characteristics of the top public DNS servers, with the most open DNS servers depending on it in Table 1. We show the number of detected clusters and the aggregate volume of servers in all clusters as a percentage of the total (refer to covered rate). The clusters of the top 4 public DNS cover over 47% of the open DNS servers. Among the 28 public DNS providers we investigated, Google is in the absolute lead, providing DNS service for 27.99% of the open DNS servers with 268 different ODNS clusters worldwide. The second place is Cloudflare, with 9.76% open DNS servers having dependencies on it.

***Unbalanced Cluster behind public DNS anycast.*** We further analyze the cluster size behind public DNS. Public DNS servers usually belong to multiple clusters distributed in different regions. This is because public DNS servers typically consist of numerous Points-of-Presence (PoPs) [37, 41] with anycast addresses. A benefit

(86% in China and 88% in South Korea) are part of clusters with sizes exceeding 100, highlighting the widespread use of forwarders depending on prominent public DNS servers such as 114DNS in China. We find even more extreme cases (*e.g.,* Bangladesh), nearly all open DNS servers fall into ODNS clusters larger than 100. These countries tend to have more servers with concerns caused by clusters. In contrast, open DNS servers in countries like the United States tend to operate within numerous medium-sized clusters. Countries like Germany and France exhibit many clusters consisting of only one server, suggesting a higher prevalence of individual resolvers. This means less cross-dependency between servers, thereby mitigating the risk of single points of failure.

> ***Observation 2:*** There is a significant difference in the distribution of cluster size across countries, where some (*e.g.,* CN) are dominated by large-size clusters (*i.e.,* ≥ 100), while others (*e.g.,* DE and FR) have many clusters containing only one server. This implies different DNS infrastructures.

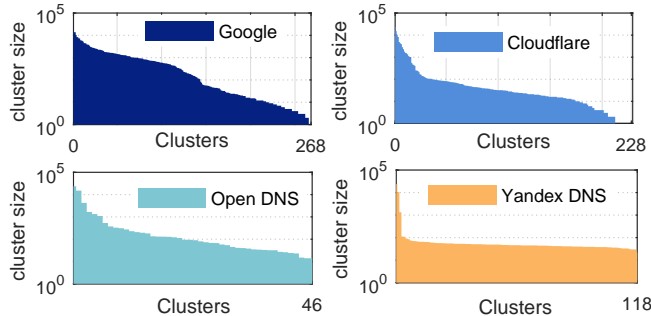

**Figure 8: Size of clusters for public DNS.**

**Table 2: Details of Top10 Clusters for Google public DNS and the countries that most servers are mapped to.**

| Country | cluster size | Country | cluster size |
|---------|-------------|---------|-------------|
| CN | 23482 | ZA | 8223 |
| UA | 13810 | BD | 7346 |
| ID | 13806 | ID | 7229 |
| RU | 10432 | BD | 6008 |
| BD | 8356 | BD | 5879 |

of our methodology is that this structure can be discovered, *i.e.,* the same public DNS address can belong to different clusters when using vantage points from different regions.

However, the population of forwarders in clusters of different PoPs is unbalanced, which may violate the original design intention for facilitating load balancing and attack defense [32]. We show the size of clusters of popular public DNS in Fig. 8 in descending order (including Google Public DNS, Cloudflare, OpenDNS, and Yandex DNS). The maximum cluster size exceeds 23k, indicating that a small subset of PoPs with large clusters in public DNS infrastructures may experience more concentrated traffic from a higher number of forwarders. Since DNS forwarders can serve as potential vectors for DDoS attacks [7], PoPs with larger clusters are exposed to more forwarders and thus face a higher risk of DDoS attacks compared to those with smaller clusters [22].

Specifically, we detail the top 10 clusters by size for Google Public DNS in Table 2. Our results reveal that large clusters are concentrated in certain regions (e.g., CN, UA, ID), indicating a high number of forwarders in these areas, but potentially limited PoPs were deployed. Such deployment may result in some PoPs facing greater pressure than those in more established areas (*e.g.,* We found 23 clusters located in the US, with an average size of only 637).

> **Observation 3:** Popular public resolvers lead the clusters that consist of a large portion of open resolvers, showing another evidence of concentration. The use of anycast results in many clusters for one public DNS, where clusters are constituted by unbalanced server populations.

**Table 3: Types of IP in problematic responses.**

| Response type | # of servers | # of clusters |
|---------------|-------------|---------------|
| Public address | 87,753 | 7,057 |
| Private address | 1,473 | 15 |
| Loopback address | 2,177 | 274 |

## 4.4 Problematic Clusters: Misconfiguration and Maliciousness

In our measurements, not all ODNS servers direct us to the correct destination. We found a total of 91,403 (9.3 %) open DNS servers in 7,346 ODNS clusters that responded to an unexpected A record to us (which does not come from our ADNS). Using these ODNS servers for DNS services during web access could lead to unpredictable and potentially harmful outcomes.

***What type of records are in these responses?*** Table 3 presents the types of IP addresses included in these problematic response records. The problematic A records comprise loopback addresses (*e.g.,* 127.0.0.1) and private addresses (*e.g.,* 10.0.0.1). A more dominant case (from 87,753 servers) involves the ODNS servers returning a third-party public IP, which redirects our web request to a specific, potentially unintended destination. These unexpected responses may be due to special configurations set by administrators or the result of malicious resolution behavior.

***How do clusters amplify the problematic responses?*** Such misconfigurations will lead to all forwarders within the ODNS cluster returning the same incorrect answer and causing a cluster-level failure. We then investigate the impact scope of problematic clusters by analyzing the distribution of cluster size and geolocations. As shown in Fig. 9(a), the clusters responding with loopback address and private address mainly consist of a single server. In contrast, the impact of unexpected public addresses is more likely to be amplified by ODNS clusters since the vast majority of clusters have a size of 2-100. This means that the same error responses from the upstream node will have an amplified effect (impact on more than one server) at the cluster level. Fig. 9(b) illustrates the top 5 countries with the most problematic clusters (response public addresses), note that nearly four thousand problematic clusters are located in the US, with over 80,000 ODNS servers responding with incorrect answers.

***Where do these responses lead us to?*** To explore the origins and purpose of such clusters, we performed reverse DNS lookup to identify the domains associated with public addresses in problematic responses, as shown in Table 4. The majority direct us to hosts of cloud hosting providers like HostGator and Bluehost, where tenants may inadvertently set up misconfigured DNS servers.

We send HTTP requests to these IPs to evaluate the types of pages that these IP addresses lead us to. We show the Responses in Table 5. Among the successful responses, most pointed to parked domains (218 instances), suggesting that these domains may not be actively used or are reserved for advertising purposes. In redirection responses, 166 IPs returned a redirect response code and we found IPs from 65 clusters (63 in VG, 1 in DE, and 1 in US, affecting 1581 ODNS servers) led to pages highly related to malicious, phishing, or abused parked domains (*e.g.,* securesearchnow [21, 28, 39]).

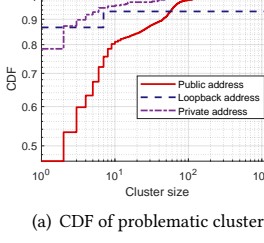

(a) CDF of problematic clusters.

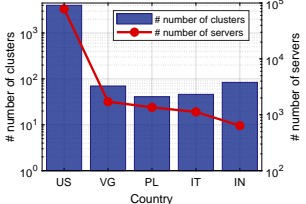

(b) Top 5 countries with problematic clusters (left) and affected servers (right).

**Figure 9: Problematic cluster size and their geolocation.**

**Table 4: Top SLDs associated with unexpected records.**

| SLD | # of servers | # of clusters |
|---|---|---|
| hostgator | 33,488 | 982 |
| bluehost | 29,994 | 1,411 |
| hostmonster | 4,616 | 115 |
| justhost | 3,148 | 78 |
| seohost-mail | 1,308 | 1 |
| flashstart | 1,069 | 1 |
| accountservergroup | 913 | 119 |
| fortinet | 734 | 1 |
| webhostbox | 534 | 43 |

**Table 5: Response type statistics.**

| Response Type | Subcategory | # of clusters |
|---|---|---|
| **Success** | Parked Domain | 218 |
| | Filtered/Blocked | 60 |
| | Error Page | 72 |
| | Others | 29 |
| **Redirection** | Malicious | 65 |
| | Normal | 101 |
| **No Response** | — | 3648 |
| **Error** | — | 2864 |

> ***Observation 4:*** Some DNS servers exhibit misconfigurations or malicious behavior. ODNS clusters amplify the impact of these problematic responses (*e.g.,* 65 problematic clusters affected 1581 ODNS servers), leading to cluster-level security vulnerabilities and misrouting of web traffic to potentially harmful destinations.

# 5 ANALYSIS INSIDE CLUSTERS

In the ODNS infrastructure, the cache behavior of forwarders speeds up the response procedure and reduces the overhead of upstream resolvers. Given the multi-layer forwarding dependency inside ODNS clusters, it is critical to give a deeper insight into how requests reach

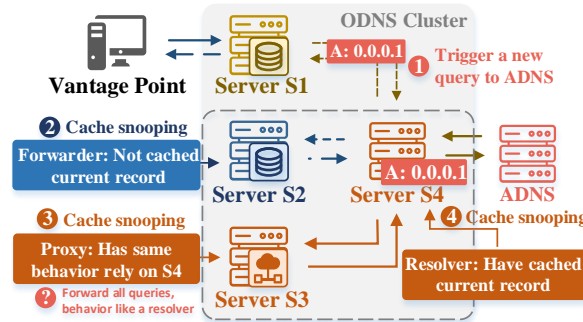

**Figure 10: Server classification inside ODNS cluster.**

upstream servers and how forwarders cache the responses. In this section, we analyze the diverse caching behaviors of forwarders inside clusters as well as the potential security risks.

## 5.1 In-cluster Server Classification

**Challenges:** Dissecting the forwarding and caching behavior of servers within ODNS clusters is non-trivial, as the forwarding process between the forwarder and the upstream server is unknown to clients (*i.e.,* we don't know whether the responses obtained from the forwarder's own cache or is forwarded directly upstream).

**Methodology:** The forwarder's caching behavior can be measured by leveraging the forwarding chain inside the cluster: when a downstream forwarder receives a new response, the record of the response is cached by all upstream servers along the path. As shown in Fig. 10, we leverage caching behaviors in the forwarding chain to analyze server caching types. First, our vantage point sends A queries for a controlled domain to a randomly selected server in the ODNS cluster (①). The query propagates through the forwarding chain, while the response caches along the way. Next, we do cache snooping [3] for all other servers in the cluster (②-④).

**Results analysis:** With the cache-snooping results, we classify all servers into the following types based on their behavior:

- **Caching forwarders**: Servers *have not cached* current record is a *caching forwarder* ($S2$), indicating that this server is not an upstream resolver. Instead, it is a forwarder and has independent cache processing itself.

- **Proxies & resolver**: Servers *have cached* current record is a *upstream resolver* ($S4$) or *non-caching proxy* ($S3$). This signifies that the cache record passed through such servers when the DNS query was initially performed. While non-caching proxies [10] simply follow the status of the upstream resolver, such proxies are classified into the same category with upstream resolver.

- **New-trigger**: In addition, if our cache detection triggers a new DNS response from ADNS, we mark such server as a *new-trigger*.

## 5.2 Results and Analysis

We conduct measurements of all clusters we got in ODNS clustering. The measurement results of each cluster are presented in Fig. 11.

---
[3]We use response time-based method [17] since it has no prior requirements for forwarder configuration [15, 24].

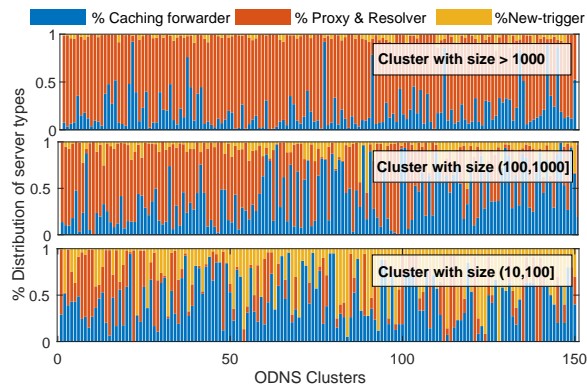

**Figure 11: Cluster composition analysis results.**

For illustration purposes, we randomly selected 150 clusters across three different size ranges without losing generality.

***Classification of servers:*** We found that a large portion of the ODNS address space is composed of proxies or upstream resolvers, which account for more than 61.7% of all detected ODNS servers. Since there are only a few upstream resolvers in a single DNS cluster due to our clustering methodology, the servers in this category are mainly proxies. From the perspective of clusters, we found a more extreme case that the proportion of proxies in 131 clusters exceeded 95%. This means that although the servers in these clusters show a two-layer structure of forwarder-resolver, they do not form a multi-level cache, and the DNS query traffic from the client will still directly reach the upstream resolver.

***Potential risks with proxies:*** Although assigning non-caching proxies to an upstream server is allowed by previous standard [10] and public DNS providers [3], it poses a security risk. Each DNS query sent to such forwarders directly reaches the upstream server, making these open forwarders vulnerable to being exploited for DDoS attacks. We recommend implementing appropriate access controls when configuring DNS forwarding devices, rather than processing all network-wide requests indiscriminately.

***In-cluster server composition pattern:*** From Fig. 11, we can intuitively observe a clear trend in the proportion of server types within clusters of different sizes. Notably, larger clusters tend to have a higher proportion of non-caching proxies since the upstream DNS server of proxies is typically a public DNS server or one assigned automatically by the ISP, which results in a larger cluster size. As a result, large clusters often face a greater risk of traffic concentration from proxies. In addition, the proportion of new-trigger servers is higher in smaller-size clusters, indicating such servers may no longer be part of the current cluster due to configuration changes.

> ***Observation 5:*** ODNS cluster consists of forwarders with diverse behaviors. Notably, the majority of forwarders (61.7%) in large clusters are non-caching proxies, which may lead to increased DNS traffic pressure on upstream resolvers and in turn serve as potential entry points for attackers.

# 6 DISCUSSION

## 6.1 Implications of ODNS Clusters

Based on our results and observations, client-side dependencies may give rise to potential risks.

***Single points failure caused by cluster.*** DNS queries from different forwarders will be concentrated on a small set of resolvers. Such concentration caused by ODNS cluster raises the risk of single-point failure [12, 26, 31]. Configuring primary and secondary DNS servers that have no dependencies (*i.e.,* exist in different ODNS clusters) can help mitigate the risk of single-point failure.

***Cluster-level malicious behavior.*** The size of the ODNS cluster reveals the potential attack scope of DNS infrastructure under malicious behavior, and this risk has regional differences because of the disparity in terms of cluster size distribution across countries. Clusters containing a large volume of servers may amplify the impact of malicious behaviors [47]. For example, hijacking [19] or cache-poisoning [48] against key nodes (*e.g.,* the egress resolver) in a large ODNS cluster can affect a wide range of (unseen) forwarders. Deploying DNS infrastructure as multiple small-sized ODNS clusters without dependencies helps reduce the effect of malicious behavior.

***Exploited forwarders behind resolvers.*** As DNS forwarders can potentially function as attack vectors for DDoS attacks [7], resolvers with more forwarders facing higher DDoS risks [22]. The ODNS clustering result can effectively assistant estimating the number of forwarders behind each resolver and their types, which helps assess potential risks to DNS infrastructure. Meanwhile, we strongly recommend stricter access control when deploying DNS forwarders.

## 6.2 Limitation and Ethics

We leverage transparent forwarders [33] as additional vantage points in our measurements in order to improve the coverage for popular public DNS (*e.g.,* Google DNS). However, the transparent forwarders can only measure the public DNS specified in their forwarding strategy, upstream resolvers that are not widely used by transparent forwarders may not be fully measured (*e.g.,* 114DNS). We will further increase the number of our vantage points and achieve finer-grained regional coverage in the future.

For ethical considerations, all measurements were conducted with our own controlled domain names, thereby preventing any query traffic from reaching third-party authoritative servers. Our methodology only needs to send 5 A-type queries per open DNS server in each controlled vantage point in a round of clustering, thereby avoiding disruption to normal services.

# 7 CONCLUSION

In this paper, we introduce the concept of ODNS clusters and propose methodologies for measuring the dependence between open DNS servers. We conducted measurements for 1.9 million open DNS servers in the wild and formed 81,636 ODNS clusters. Measurement results show the ODNS infrastructure differences crossing regions and the network centralization led by major public DNS providers. We further discuss the potential risks of problematic clusters and non-caching proxies with a cluster-level perspective. Our findings shed light on the current status of the ODNS ecosystem, and possible enhancements to the ODNS infrastructure.

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
