# OpenReview forum: "ODNS Clustering: Unveiling Client-side Dependency in Open DNS Infrastructure"
_ACM.org/TheWebConf/2025/Conference — WWW 2025 Poster_

### Official Review · Reviewer_hhJ2 · 2024-11-18

**Novelty:** 5
**Technical Quality:** 5

**Review:**

Thank you for your submission. Active ADNS-based measurement shows immense promise for complete characterization of the DNS ecosystem, and I enjoyed reading about its applications in the measurement of ODNS clustering. The methodology is clearly explained, and the study results have clear security impacts that are succinctly described (S2.2). I take issues with the overall validity of some results, but in general find the work offers an interesting characterization of ODNS that could be augmented by future work.

# Strengths:
* Methodology is well-described and intuitive. S3.1 describes the general approach, and S3.2 then shows a natural expansion of this to the real-world noisy setting.
* Vantage point is well-described. The use of transparent forwarders to augment the physical vantage point is clever and well-founded.

# Weaknesses:
* Some claims based on raw numerical results are not necessarily justified. For instance, is 82k clusters out of 1.9M servers an unexpectedly-high amount of concentration? Especially given servers are discovered across all IPv4s, it is not particularly surprising that many of these are, e.g., filtering DNS proxies in front of some popular infrastructure. See also below on S4.4.(Amplifying Responses)
* The motivations behind measured ODNS servers are not clear. For instance, discussion in 4.4 implies that many of these may be poorly-configured ADNS servers.

# Other comments
*4.2*: The original hypothesis here relates to server placement for latency, but results refer to country codes. I would be curious if this is skewed by geographic size of each country, and if approximate distance between servers (admittedly a metric with worse available data) might account for country-level differences.

*4.4: amplifying responses*: This claim could be softened, since an amplification in the number of resolvers giving problematic responses does not necessarily mean users are affected. Is it possible to give some sense of which ODNS servers are actually in use? For instance, are they associated with some ISP with likely users?

The data following ("Where do these responses lead us to") seems to imply that many of these ODNS servers may in fact be pooly-configured ADNS servers. Table 4 in particular seems to imply this.

*Fig.11*: You could consider sorting these DNS servers by one of the series to effectively present an eCDF. Right now the bar-chart approach makes it quite difficult to reason about the distribution. Further, it does not appear this figure actually depicts percentages. Consider referring to these as proportions or ratios.

**Questions:**

Did you take steps to ensure that identified ODNS servers were functioning as public resolvers on purpose, as opposed to being poorly-configured ADNS servers?

**Reviewer Confidence:**

3: The reviewer is confident but not certain that the evaluation is correct

**Scope:**

3: The work is somewhat relevant to the Web and to the track, and is of narrow interest to a sub-community

---

### Official Review · Reviewer_Vu5U · 2024-11-25

**Novelty:** 4
**Technical Quality:** 3

**Review:**

This paper introduces the concept of ODNS Clusters, representing dependencies among open DNS servers, and unveils the implications of these clusters on DNS infrastructure, including security risks and failure points. Through rigorous methodology, it identifies 81,636 ODNS clusters from 1.9 million open DNS servers and provides key insights, such as the concentration of public DNS providers and regional disparities in DNS infrastructure. Besides, The paper further discusses the risks associated with misconfigurations and malicious behaviors, emphasizing the amplified effects within ODNS clusters.

**Weaknesses:**

1. While this paper proposes a new ODNS clustering methodology, it doesn’t compare it with existing DNS clustering or dependency detection methods.

2. The paper uses a simple method for server classification based on cache behavior. However, this method does not account for transient caching states or TTL-based cache expirations that might affect the classification outcome.

3. There are inconsistencies in the use of key terms, such as "Client-side" and "client side", "ODNS Cluster" and "DNS Cluster" which are used interchangeably. Additionally, abbreviations such as ODNS and TLD are introduced without providing their full forms initially, which needs to be paid attention to.

**Questions:**

1. You mention that a threshold value of α=0.1 was used for cluster aggregation. Did you experiment with different values of α, and if so, how did it impact the clustering results?

2. In Section 4.4, you have identified 91,403 servers with unexpected responses. Are there any identifiable patterns (e.g., geographic or infrastructural) that could help explain why these servers exhibit such behavior?

3. The server classification method relies heavily on cache snooping. How does this method handle variability in cache states due to TTL expiration? Would repeated measurements over time yield consistent classification results?

4. In Step 3 of Section 3.1, you mention using shared A records to group servers into clusters. However, the criteria for defining "dependency" between servers are not clearly explained. Firstly, I hope to know how you determine if two servers should be clustered based on the shared A records. Is there a minimum number of shared responses that define dependency? Secondly, how do you handle false positives where servers might have coincidental matching responses but are not functionally dependent?

**Reviewer Confidence:**

3: The reviewer is confident but not certain that the evaluation is correct

**Scope:**

3: The work is somewhat relevant to the Web and to the track, and is of narrow interest to a sub-community

---

### Official Review · Reviewer_gH6n · 2024-11-28

**Novelty:** 3
**Technical Quality:** 3

**Review:**

This paper introduces the concept of ODNS clusters and provides a new methodology for measuring client-side dependencies in DNS infrastructure, which is a significant contribution to the field. The paper relies heavily on data from major public DNS providers, which might not capture the full diversity of DNS infrastructures, especially in regions with less reliance on these providers.

**Questions:**

Q1. Could you provide more details on how your sampling strategy ensures representative coverage across different regions, particularly in areas with less reliance on major public DNS providers?

Q2. How have you validated the stability of the ODNS clusters over time? Are there any measures in place to account for dynamic changes in DNS server configurations and dependencies?

**Reviewer Confidence:**

4: The reviewer is certain that the evaluation is correct and very familiar with the relevant literature

**Scope:**

3: The work is somewhat relevant to the Web and to the track, and is of narrow interest to a sub-community

---

### Official Review · Reviewer_PP2s · 2024-11-28

**Novelty:** 6
**Technical Quality:** 6

**Review:**

Good points:
This is a well written paper with good quality illustrations and clear explanation of their set up.
The problem is an interesting one which needs study.
Thorough measurement and experimentation clearly described.

Bad points:
I am not convinced the algorithm 1 is "deterministic" in a sense described below. EDIT: This was addressed.


It is encouraging to see authors take such trouble to clearly explain their setting and methods.

In fig 5 (b) detail of the high cluster sizes is lost as the CDF approaches 1 for sizes > 100. A logscale y can help this -- but it might be harder to interpret.

I find algorithm 1 a bit unncessary as a description. The concept it is describing is simple but algorithm 1 doesn't add much. The point of an algorithm in a paper is to make clear something that was hard to understand written in English. The algorithm simply works through the concepts in slightly long winded code. Not a huge problem at all, in a way it is a complement to the clarity of the original description.

The algorithm merges two sets if there is an overlap. But I *think* I may be wrong the algorithm outcome is determined by some random factors. The parameter alpha determines whether two sets should merge. Let us imagine set A matches with set B and set C according to that criteria. Set A merges with set B because set B happens to be earlier in the search order. It is not clear to me that the combined set A+B will merge with C as set A+B may no longer have such a good match. If I am right then the outcome will change depending on how the sets are ordered as processed.
This is not necessarily a problem, a lot of algorithms have this property -- for example, the Louvain algorithm for partitioning a network into communities. However, I would want to know if the conclusions are similar if the ordering was different. One way to test would be to randomise the order in which the clusters were processed.


Table 2 might be helpful to include the actual country name. I'm normally good with TLD but BA and UA defeated me (I could have guessed but I shouldn't have to).

**Questions:**

If I am correct the outcome of the cluster algorithm depends on the order of its inputs. (A) is that true (B) if it is do your conclusions still hold if the order of its inputs is randomised. [It may be this experiment is not necessary but it seems a simple experiment.]

Can we get any intuition why the countries in table 2 are chosen? Large countries? Countries with many AS to cluster? Countries with infrastructure developed more recently?

(These questions are addressed.)

**Reviewer Confidence:**

3: The reviewer is confident but not certain that the evaluation is correct

**Scope:**

4: The work is relevant to the Web and to the track, and is of broad interest to the community

---

### Official Review · Reviewer_qUHA · 2024-11-30

**Novelty:** 6
**Technical Quality:** 5

**Review:**

This paper presents a robust analysis of ODNS clusters, demonstrating a clear and systematic approach to analyzing dependencies and vulnerabilities in DNS infrastructures. The methodology is well-structured, with innovative use of controlled vantage points and transparent forwarders to measure DNS dependencies. However, the reliance on transparent forwarders introduces some limitations, particularly in capturing less common DNS resolvers. The paper is also generally clear and well-organized, with sections logically divided into methodology, results, analysis, and implications. In particular, the figures effectively complement the text. This work introduces the novel concept of ODNS clusters in DNS infrastructure, demonstrating originality and clearly highlighting issues such as DNS centralization, cluster misconfigurations, and proxy behavior.

Pros
- A well-written paper with appropriate measurement and analysis.
- Comprehensive measurement of 1.9 million open DNS servers.
- Introduces ODNS clusters to study DNS dependencies in a new light.
- Detailed statistical and geographical analyses highlighting the DNS infrastructure's regional disparities.
- The study ensures no disruption to third-party servers by exclusively using controlled domains.


Cons
- Transparent forwarders may not capture all upstream resolvers, which limits the scope and completeness of the study.
- The regional disparities identified warrant deeper exploration of their socio-economic and policy implications.
- The concept of ODNS clusters, while novel, builds upon prior research on DNS dependencies and transparency, lacking a detailed justification for why it should be introduced as a distinct framework.

**Questions:**

Transparent forwarders only cover widely used upstream resolvers. How would the results change if more diverse vantage points were incorporated to include less frequently used upstream resolvers? Do you have plans to address this limitation by incorporating more diverse vantage points in future work?

There are significant regional disparities in cluster size and DNS infrastructure. What socio-economic or policy factors do you think contribute to these disparities, and how could they be mitigated?

The concept of ODNS clusters is novel. How do you envision this concept applying to practical applications, such as DNS monitoring, threat detection, or infrastructure optimization?

The vantage points were concentrated in specific regions. Could this geographical bias, due to uneven distribution of vantage points, have affected the accuracy of the clustering results, particularly in regions with less coverage?

Are ODNS clusters the only way to move beyond the limitations of previous studies (e.g., cache hit issues)?

**Reviewer Confidence:**

2: The reviewer is willing to defend the evaluation, but it is likely that the reviewer did not understand parts of the paper

**Scope:**

3: The work is somewhat relevant to the Web and to the track, and is of narrow interest to a sub-community